behaviour, ecology

energy landscape, migration, soaring, bio-logging, route selection, movement ecology

**Author for correspondence:**
Elham Nourani
e-mails: enourani@ab.mpg.de, mahle68@gmail.com

# The interplay of wind and uplift facilitates over-water flight in facultative soaring birds

Elham Nourani[1,2], Gil Bohrer[3], Paolo Becciu[1,2,4,5], Richard O. Bierregaard[6], Olivier Duriez[7], Jordi Figuerola[8], Laura Gangoso[8,9,10], Sinos Giokas[11], Hiroyoshi Higuchi[12], Christina Kassara[11], Olga Kulikova[2,13], Nicolas Lecomte[14], Flavio Monti[15], Ivan Pokrovsky[1,13,16], Andrea Sforzi[17], Jean-François Therrien[18], Nikos Tsiopelas[19], Wouter M. G. Vansteelant[8,10], Duarte S. Viana[20,21], Noriyuki M. Yamaguchi[22,23], Martin Wikelski[1,2,24] and Kamran Safi[1,2]

[1]Department of Migration, Max Planck Institute of Animal Behavior, Germany
[2]Department of Biology, University of Konstanz, Germany
[3]Department of Civil, Environmental and Geodetic Engineering, The Ohio State University, USA
[4]Department of Evolutionary and Environmental Biology, and Institute of Evolution, University of Haifa, Israel
[5]Department of Ecology and Evolution, University of Lausanne, Switzerland
[6]Ornithology Department, The Academy of Natural Sciences of Drexel University, USA
[7]Centre for Evolutionary and Functional Ecology, Montpellier University-CNRS, France
[8]Department of Wetland Ecology, Estación Biológica de Doñana, Spain
[9]Department of Biodiversity, Ecology and Evolution, Faculty of Biology, Complutense University of Madrid, Spain
[10]Institute for Biodiversity and Ecosystem Dynamics (IBED), University of Amsterdam, The Netherlands
[11]Department of Biology, University of Patras, Greece
[12]Research and Education Centre for Natural Sciences, Keio University, Japan
[13]Russian Academy of Sciences, Institute of the Biological Problems of the North, Russia
[14]Canada Research Chair in Polar and Boreal Ecology, Department of Biology, University of Moncton, Canada
[15]Department of Physical Sciences, Earth and Environment, University of Siena, Italy
[16]Institute of Plant and Animal Ecology, Russian Academy of Sciences, Russia
[17]Maremma Natural History Museum, Italy
[18]Hawk Mountain Sanctuary, Acopian Center for Conservation Learning, USA
[19]Hellenic Ornithological Society/BirdLife, Greece
[20]German Center for Integrative Biodiversity Research (iDiv) Halle-Jena-Leipzig, Germany
[21]Leipzig University, Germany
[22]Graduate School of Fisheries and Environmental Sciences, Nagasaki University, Japan
[23]Organization for Marine Science and Technology, Nagasaki University, Japan
[24]Centre for the Advanced Study of Collective Behaviour, University of Konstanz, Germany

EN, 0000-0003-4420-3902; GB, 0000-0002-9209-9540; PB, 0000-0003-2145-6667; OD, 0000-0003-1868-9750; JF, 0000-0002-4664-9011; LG, 0000-0002-6205-6769; SG, 0000-0003-2809-1605; HH, 0000-0002-1215-2736; CK, 0000-0002-4824-0674; OK, 0000-0002-4029-9452; NL, 0000-0002-8473-5375; FM, 0000-0001-8835-1021; IP, 0000-0002-6533-674X; AS, 0000-0002-7640-084X; J-FT, 0000-0002-1046-0962; NT, 0000-0002-2896-6144; WMGV, 0000-0002-9447-8587; DSV, 0000-0002-7864-0871; NMY, 0000-0001-6188-5744; KS, 0000-0002-8418-6759

Flying over the open sea is energetically costly for terrestrial birds. Despite this, over-water journeys of many birds, sometimes hundreds of kilometres long, are uncovered by bio-logging technology. To understand how these birds afford their flights over the open sea, we investigated the role of atmospheric conditions, specifically wind and uplift, in subsidizing over-water flight at a global scale. We first established that $\Delta T$, the temperature difference between sea surface and air, is a meaningful proxy for uplift over water. Using this proxy, we showed that the spatio-temporal patterns of sea-crossing in terrestrial migratory birds are associated with favourable uplift conditions. We then analysed route selection over the open sea for five facultative soaring species, representative of all major migratory flyways. The birds maximized wind support when selecting their sea-crossing routes and selected greater uplift when suitable wind support was available. They also preferred routes with low long-term uncertainty in wind conditions. Our findings suggest

that, in addition to wind, uplift may play a key role in the energy seascape for bird migration that in turn determines strategies and associated costs for birds crossing ecological barriers such as the open sea.

## 1. Introduction

Dynamic atmospheric conditions largely define the energetic costs of flight for birds. Tail winds, for example, permit birds to reduce air speed while maintaining the speed of travel, helping them to save energy [1,2]. Likewise, rising air as a consequence of warm air columns known as thermals and orographic uplift created by the interplay between wind and topography can push flying animals upwards and reduce the energetic costs of remaining airborne [3–5]. However, the energy availability landscape [6] is interspersed with patches where energetic subsidies in the atmosphere are weak or absent, impeding efficient movement. Flight over these areas becomes energetically costly, yet some animals regularly engage in such seemingly risky flights, particularly during migration [7,8]. How birds afford their flights across migratory barriers remains an open and important question for understanding the evolution of migratory routes and sea-crossing strategies.

The open sea is considered a major migratory barrier for all terrestrial species, and particularly for soaring birds [8,9]. This is rooted in observational studies of birds gathering in large numbers at bottlenecks prior to setting out over even relatively short over-water passages [8,10]. The existence of such large aggregations, together with the technological difficulty of observing birds over the open sea, led to a general understanding that sea-crossing poses a formidable challenge for birds that are unable to land or forage at sea [11]. Some early studies looked beyond the role of geography alone and suggested that the extent to which terrestrial birds aggregate through overland flyways vary greatly across space and time [12–14]. They further suggested that weather conditions play an important role in terrestrial birds' ability to embark on, and sustain, over-water flight [15]. Advances in bio-logging technology have since created a clearer picture of sea-crossing behaviour in terrestrial birds.

Bio-logging has confirmed extremely long sea-crossings in terrestrial birds [16,17]. It has provided evidence that the prevalence and extent of this behaviour varies according to flyway [8,11], season [18,19] and morphology [20]. We now have ample evidence that, in most cases of sea-crossings, atmospheric support, mostly in the form of wind support (i.e. the length of the wind vector in a bird's flight direction) is an important facilitator [18,21–24].

An emerging hypothesis is that uplift also plays a role in the energy seascape for soaring bird migration. Uplift can reduce the energetic costs of remaining airborne. Additionally, soaring birds can take advantage of strong uplift to soar. Past bio-logging studies measured flight altitude in soaring birds to provide indirect proof of thermal soaring behaviour at sea [25,26]. More recently, high-resolution GPS tracking documented the circling flight pattern and vertical aerial climb of migrating ospreys over the Mediterranean Sea [27]. Duriez *et al.* [27] also confirmed the earlier suggestions that $\Delta T$, defined as the difference in temperature between the sea surface and the air, can be used as a proxy

for uplift potential over water [28]. Positive $\Delta T$ values correspond to upward moving air (warmer sea surface than air), while negative values can be interpreted as sinking air, termed subsidence. This proxy was consequently adopted to quantify the energy seascapes that enable juvenile European honey buzzards to survive longer sea-crossings compared to their earlier migrating adult conspecifics [19]. Yet, whether $\Delta T$ is a meaningful correlate of upward moving air has not been quantitatively tested.

In this study, we investigate sea-crossing behaviour at the global scale to assess the role of uplift and wind in shaping the energy seascapes for terrestrial bird migration. To do this, we use bio-logging data collected for five raptor species that perform long sea-crossing journeys. These species differ in size, morphology and stop-over strategies and their overall dependence on soaring flight to cover long distances. However, they are all facultative soaring migrants and are expected to preferentially use soaring over flapping flight whenever possible, including flights over open water. This provides an opportunity to investigate sea-crossing behaviour at all major migratory flyways at spatio-temporal scales equivalent to that of the publicly available global atmospheric information. Using these data, we set out to establish whether $\Delta T$ is a meaningful proxy for uplift potential over water, by testing its relationship with convective velocity ($w^*$) [29]. We further hypothesize that (i) sea-crossings are associated with temperature gradients that are indicative of uplift in all flyways and (ii) both wind and $\Delta T$ influence over-water route selection.

## 2. Methods

### (a) Bio-logging dataset

We compiled a bio-logging dataset containing migratory trajectories of birds that regularly perform sea-crossing during autumn. We did not include spring migration in this study, due to the limited amount of data for spring migration compared to autumn. Our dataset was comprised five species: the Oriental honey buzzard *Pernis ptilorhynchus* and the grey-faced buzzard *Butastur indicus* in the East Asian flyways, the osprey *Pandion haliaetus* and the peregrine falcon *Falco peregrinus*, in both the African-Eurasian and the American flyways, and the Eleonora's falcon *F. eleonorae* in the African-Eurasian flyway. These birds are all facultative soaring birds. Their dependence on uplift varies, with the falcons and the osprey being less dependent on uplift than the buzzards [11].

We focused only on sea-crossing behaviour during migration to ensure a common flight purpose among all species and individuals in the analyses. We only included adults as they actively select their route based on experience, unlike juveniles that follow an innate direction of migration, probably without established route selection criteria [30,31]. We limited our analysis to sea-crossing trips longer than 30 km, which corresponded to the spatial resolution of our environmental data (see Route selection analysis below).

### (b) $\Delta T$ and convective velocity

To determine whether $\Delta T$ is a meaningful measure of uplift velocity, we estimated the relationship between this variable and the convective velocity scale, $w^*$ [29]. $w^*$ is estimated on the basis that the buoyancy associated with surface heat flux produces uplift:

$$w^* = \left[ gz \left( \frac{(w'T') + 0.61 T_2 m(w'q')}{\theta} \right) \right]^{1/3},$$

where $g$ [m s$^{-2}$] is the gravitational acceleration, $z$ [m above reference sea level] is the depth of the atmospheric boundary layer, the term $(w'T' + 0.61\ T_2m(w'q'))$ approximates the surface virtual potential temperature flux, where $(w'T')$ [K m$^{-2}$ s$^{-1}$] is the surface temperature flux and $(w'q')$ [g$_{water}$/kg$_{air}$ m$^{-2}$ s$^{-1}$] is the surface water vapour flux and $T_2m$ [C] is the air temperature near the sea surface. $\theta$ [K] is the virtual potential temperature at the height of the boundary layer top ($z$) and was approximated using the wet adiabatic lapse rate (typical for oceanic boundary layers) as

$$\theta = T_2m + 273.15 + 0.006z.$$

We used data from the ERA-interim reanalysis database (spatial and temporal resolution of 0.75 degrees and 3 h, respectively) provided by the European Center for Medium-Range Weather Forecast (ECMWF; https://www.ecmwf.int). We accessed these data through the EnvDATA service [32] on Movebank (www.movebank.org), to get the values for $z$, $T_2m$, $w'T'$ and $w'q'$ (with some unit conversions). Overall, we closely followed Bohrer et al. [33] to calculate $w^*$, but including the contribution of the surface water vapour flux to the surface virtual potential temperature flux, and using the wet adiabatic laps rate to estimate the potential temperature at the height of the boundary layer top.

The variable $w^*$ represents uplift and can only be calculated for positive heat fluxes. Hence, we calculated $w^*$ for all the sea-crossing points in our bio-logging dataset where $\Delta T$ was positive and estimated the relationship between the two variables using a generalized additive model.

## (c) Spatio-temporal modelling of $\Delta T$

To show the spatio-temporal variation in $\Delta T$ at the global scale, we used 40 years of temperature data. We focused on five regions where regular long-distance sea-crossing is performed by facultative soaring birds, namely South-east Asia, the Indian Ocean, the Mozambique Channel, Europe and the Americas. We downloaded sea surface temperature and temperature at 2 m above the sea for these regions for 1981–2020 from the ECMWF ERA-interim reanalysis dataset (spatial and temporal resolution of 6 h and 0.75°, respectively). We chose to use this dataset instead of the higher resolution ERA5, to reduce the time and memory needed for the analysis. We spatially filtered the data to exclude lakes, as we were only interested in the open seas and oceans. To include a proxy for the time of day, we calculated the solar elevation angle for each data point. We then created a categorical variable with three levels, night, low sun elevation and high sun elevation, corresponding to sun elevation angles below −6, between −6 and 40 degrees, and over 40 degrees, respectively.

We loosely followed Nourani et al. [19] to construct energy seascapes. In brief, we modelled $\Delta T$ as a function of latitude, longitude, day of year and time of day using the generalized additive mixed modelling (GAMM) approach. Five models were constructed in total, one per region. We extracted the timing and location of sea-crossings from our bio-logging dataset. We did not have empirical data for migration over the Indian Ocean and therefore consulted the relevant literature to extract the spatio-temporal pattern of the Amur falcon's *Falco amurensis* sea-crossing over the region [34,35]. Each model included two smoothers, one cyclic cubic regression splines smoother for the day of the year and a spline on the sphere for latitude and longitude. For both of these parameters, one smoothing curve was estimated for each level of time of day. Year was added as a random intercept in the models to control for annual variations in $\Delta T$. We also included a variance structure in the models to account for the heteroscedasticity caused by higher $\Delta T$ variance in higher latitudes. Models were fitted using the mgcv package [36] in R v. 4.0.2 [37]. We used each model to predict energy seascape maps for the autumn migration

season (August–November). We spatially interpolated the prediction rasters to a 1 km resolution for visualization purposes.

## (d) Route selection analysis

We investigated route selection by fitting a step selection function [38] to relate the probability of presence over the sea with atmospheric conditions. Every two consecutive points along a track were considered a step. Atmospheric conditions were compared between the observed step and a set of alternative steps that were available to the birds in space and time. The grey-faced buzzard was excluded from this analysis because of the low resolution of the satellite-tracking data.

We filtered our dataset to include only points over the open sea. Trajectories that intersected land (e.g. islands) were broken into sea-crossing segments. Segments shorter than 30 km, and those that included fewer than three tracking points, were removed. To ensure a uniform temporal resolution and to reduce spatio-temporal auto-correlation, we re-sampled all data to one-hourly intervals (with a tolerance of 15 min) using R package Move [39].

We prepared a stratified dataset: along with each sea-crossing segment, for each step, we generated 50 spatially alternative steps. Based on the distribution of step lengths (gamma distribution) and turning angles (von Mises distribution) estimated using over-water tracking data, fitted separately for each species-flyway combination. Diagnostics plots were used to assess the fit of the distributions. All data were then annotated using the ENV-data track annotation service [32] provided by Movebank. Each point was annotated with u (eastward) and v (northward) components of the wind, sea surface temperature and temperature at 2 m above the sea, all provided by ECMWF ERA5 reanalysis database (temporal and spatial resolution of 1 h and 0.25°, respectively). We selected the bilinear and the nearest-neighbour methods of interpolation for the wind and temperature data, respectively. We then calculated wind support [40] and $\Delta T$ using the annotated data. Additionally, to investigate whether the predictability of atmospheric conditions affected the sea-crossing route choice, we annotated each point with long-term variances (over 40 years; 1981–2020) for wind support and $\Delta T$.

We checked the annotated dataset for multicollinearity and only used variables that were not highly correlated ($r < 0.6$). Prior to analysis, we centred and scaled the predictors to mean zero and units of standard deviation (i.e. z-scores) to ensure comparability among predictors.

Step selection functions were then estimated using the integrated nested Laplace approximation (INLA) method using the INLA package [41] in R v. 4.0.2 [37]. We constructed a multilevel model with fixed effects for $\Delta T$, wind support, long-term variance of wind support and an interaction term for wind support and $\Delta T$. Species and individual IDs (nested within species) were included as random effects on the slopes. Following Muff et al. [42], we set $N(0, 10^4)$ as the prior for fixed slope parameters and penalized complexity priors $PC(3, 0.05)$ to the precisions of the random slopes. The model converged after 50 min on an Intel Core i7–8700 $12 \times 3.20$ GHz processor (running on 10 cores in parallel).

## 3. Results

We found a positive relationship between the convective velocity scale, $w^*$ and $\Delta T$ (figure 1; r2 (adj.) = 0.31; $p < 0.05$). The sea-crossing data from all species showed overnight flights (figure 1; electronic supplementary material, figure S4). There was no clear difference between the pattern of correlation between $w^*$ and $\Delta T$ in different times of the day or for different species (figure 1).

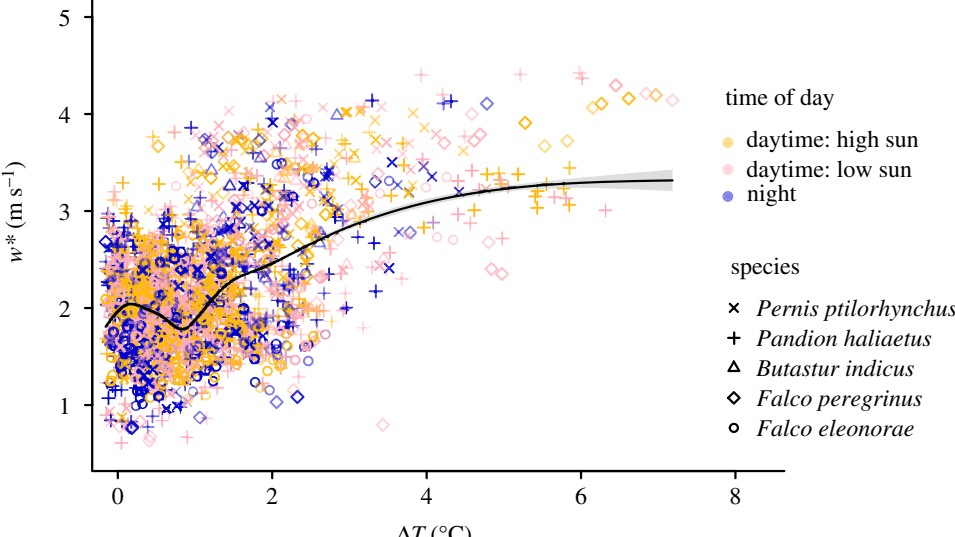

**Figure 1.** Relationship between $\Delta T$ and the convective velocity, $w^*$ (for all points with positive $\Delta T$). The sea-crossing bio-logging data (pooled for all five species) were annotated with values of $\Delta T$ and $w^*$. The smooth curve shows cubic regression splines fitted to the data predicting $w^*$ as a function of $\Delta T$ (with the 95% confidence interval). Colours and shapes represent different times of day and species, respectively. (Online version in colour.)

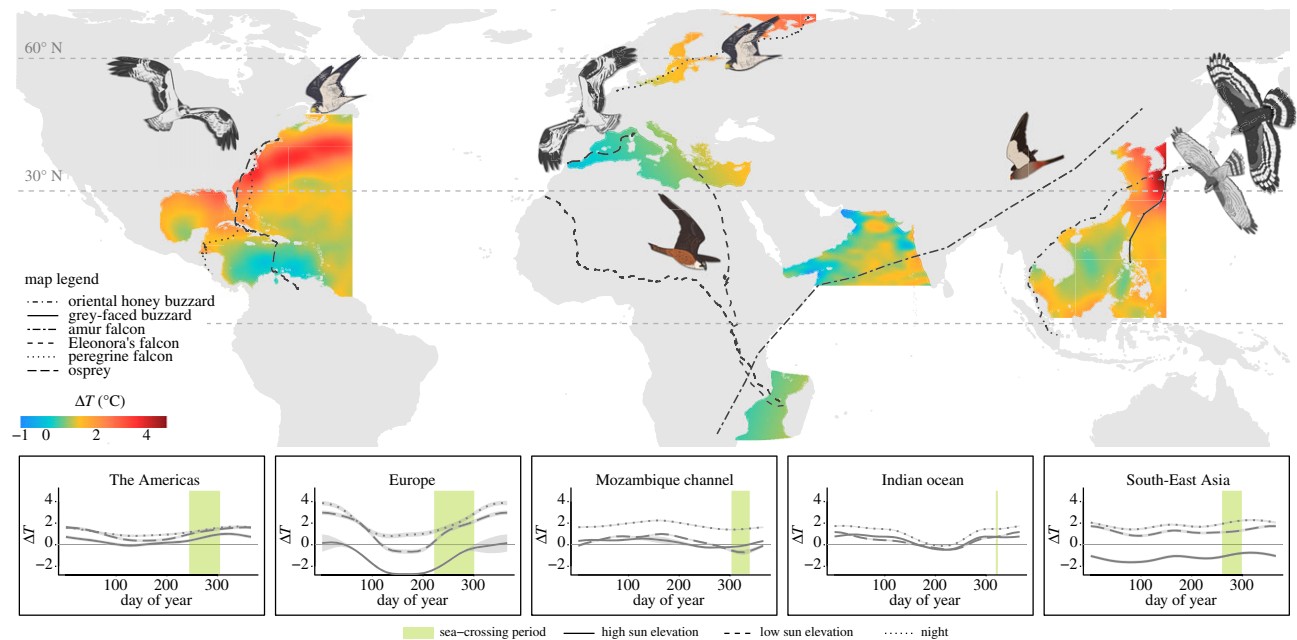

**Figure 2.** Energy seascapes for soaring bird migration in autumn. The map shows the energy seascapes for August–November derived from 40 years of temperature data. Tracks correspond to sample migratory trajectories. All tracks are based on empirical data, except for the Amur falcon, which is based on the available literature [35]. Subplots show the distribution of $\Delta T$ throughout the year in each region, for each time of day (based on summed effects from the GAMMs with 95% confidence intervals). Green shaded areas in the subplots show the timing of sea-crossing in the species flying over the corresponding region. (Online version in colour.)

The spatio-temporal pattern of sea-crossing in the six terrestrial bird species (the five species in our bio-logging dataset plus the Amur falcon) corresponded with positive uplift potential over the open sea (figure 2; see subplots for within-year and within-day variations in each region; see electronic supplementary material, table S2 for detailed GAMM outputs). The osprey was the only species flying over the open sea when the sea surface was colder than the air (i.e. negative $\Delta T$). This pattern occurred over both the Mediterranean and the Caribbean Seas (figure 2). The Eleonora's falcons flying over the Mozambique Channel also experienced relatively low and sometimes negative $\Delta T$ (figure 2). However, over-water flight in the absence of uplift did not set the two species apart from the other species in route selection regarding $\Delta T$.

We analysed over-water route selection in 112 sea-crossing tracks of 65 individuals (electronic supplementary material, table S1 and figure S3). We did not include the long-term variance of $\Delta T$ in the model, as it was correlated with the long-term variance of wind support ($r = 0.61$; $p < 0.05$). The most important variable determining over-water route selection was wind support, with a positive effect. The interaction between $\Delta T$ and wind support also showed a positive, yet smaller, effect (figure 3). The variables $\Delta T$ and the long-term variance of wind support had negative impacts on route selection. The model results suggested a greater preference for wind support in the Eleonora's falcon and a weaker preference in the osprey (electronic supplementary material, figure S1).

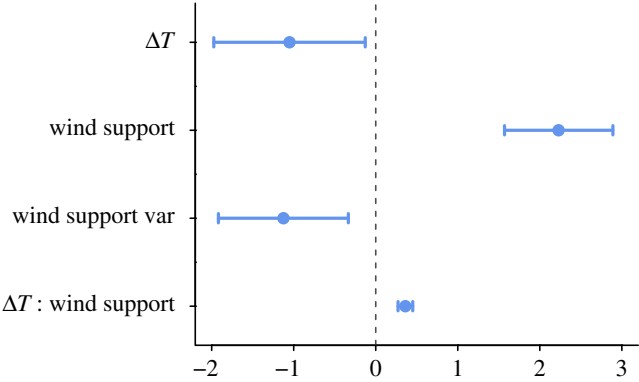

**Figure 3.** Posterior means (centred and scaled) and 95% credible intervals for the fixed effects in the INLA model. (Online version in colour.)

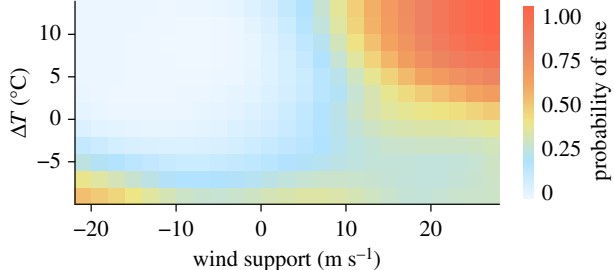

**Figure 4.** Probability of use under varying wind support and uplift conditions, based on INLA model predictions. (Online version in colour.)

## 4. Discussion

We found that, in all flyways that we studied, facultative soaring birds were more selective for wind support than uplift when flying over the open sea. This confirms previous findings that birds aim at maximizing wind support, probably to reduce the time and energy required to complete their journeys [18,21–24,43–45]. Our model also showed a positive effect of the interaction between wind support and $\Delta T$. This indicates that when wind support is favourable, the birds also select high values of $\Delta T$ to gain a better advantage of uplift (figure 4).

The selection of suitable wind support went beyond the individuals' instantaneous response to wind conditions. The birds avoided areas with high long-term variability in wind support when flying over water (figure 3). The 40-year conditions that we considered here are beyond one individual's experiences, but point to population-level preferences, which might be socially transmitted across generations to achieve efficient movement [46,47]. Reducing uncertainty in the energetic costs of migration in this way can be a mechanism for the formation of efficient migratory routes within populations.

Contrary to our expectations, after accounting for the other variables, higher $\Delta T$ values alone had a lower probability of being chosen on route selection. This suggests that in the absence of suitable wind support, the birds did not choose stronger uplifts to fuel their flights (figure 4). Still, we show that uplift conditions over the sea were generally favourable during the autumn migration season (figure 2) as well as at the start of sea-crossing (electronic supplementary material, figure S4). We speculate that the birds may avoid turbulent conditions by avoiding high $\Delta T$. The map showing mostly positive $\Delta T$ (electronic supplementary material, figures S2 and S3) indicates that even when selecting low values of $\Delta T$, the birds were still likely to

experience some amount of uplift regardless and would not prefer higher uplift for forgoing better wind support.

The sea surface tends to be warmer than the air in autumn, creating upward heat flux. This is reflected in our global energy seascape map (figure 2). The mostly positive values of $\Delta T$ on this map indicate that the spatio-temporal patterns of sea-crossing in autumn is associated with favourable uplift conditions, supporting our first hypothesis. Moreover, the range and mean of $\Delta T$ values were similar between the observed and alternative steps in our step selection function estimation (electronic supplementary material, figure S2). The birds thus faced more variability in wind conditions than in uplift, which can further explain why wind support was the most important criterion for route selection (figure 3). We further showed that $\Delta T$ has a relatively low seasonal variation, except in Europe (subplots in figure 2). As a result, birds flying along these flyways in spring could potentially benefit from uplift. However, we cannot make generalizations about sea-crossing patterns in spring based on $\Delta T$ alone. As our route selection analysis showed, wind support plays a more important role in over-water flight than uplift. Even in similar uplift conditions between the two seasons, variations in wind support can lead to loop migration patterns [48,49] and even avoidance of sea-crossing in one season [18]. Moreover, species may differ in their energy- or time-saving strategies between seasons. Uncovering the role of atmospheric subsidies in shaping sea-crossing patterns in spring requires multi-species bio-logging data, which were not available to our study.

Evidence for a soaring flight over the sea has accumulated for at least a decade. Yet, many studies overlook uplift potential when explaining sea-crossing behaviour. This could be due to the presumption that thermals do not form over the sea, as well as a lack of a reliable and easy to compute proxy for uplift over water. Studies that try to investigate uplift do so by using a variety of proxies, including air temperature [50], air temperature gradient [18], vertical air velocity [18], boundary layer height [51] and solar irradiance [52], which makes interpretation of the results and comparisons among studies difficult. $\Delta T$ is shown, by direct observations [53] and bio-logging technology [27], to be related to soaring flight. Our results confirming the correlation of $\Delta T$ with $w^*$, a widely accepted measure of uplift by the movement ecology community (so far mostly used to estimate uplift over land [33]), further point to the potential of this proxy as a measure of uplift over the open sea. Duriez *et al.* [27] showed that ospreys predictably engaged in thermal soaring over the Mediterranean Sea when $\Delta T$ values were higher than 3°C. Although we provide a range of $w^*$ values (m s$^{-1}$) for each value of $\Delta T$ in our data, the resolution of our data did not allow us to directly investigate whether and in what $\Delta T$ conditions the birds engaged in soaring flight. In fact, there remains a strong need for quantifying the amount of energy, or the realized uplift, that a bird can gain from $\Delta T$. Theoretical and high-resolution multi-sensor bio-logging studies can investigate this for birds with different morphological characteristics and under different wind conditions.

Although we show that it is possible to estimate $w^*$ using publicly available weather data, it is not an ideal proxy for measuring uplift under all circumstances over the open sea. First, $w^*$ can only be estimated for upward moving air. This means that there will be no values estimated for subsidence. $\Delta T$ does not have this limit, as it provides positive values for

uplift and negative values for subsidence. Second, calculating $w*$ requires at least a basic knowledge of meteorological concepts and units. Different reanalysis datasets provide different variables for potential and water vapour fluxes, making our method of calculating $w*$ difficult to adapt for other studies. This is while $\Delta T$ can be calculated only using two simple variables, sea surface and air temperature, and yields the same result whether temperature data are obtained in units of kelvin or Celsius. It is therefore a more approachable and possibly reliable measure than $w*$. Lastly, although it is ideal to be able to calculate uplift in meters per second, $w*$ has not been ground-truthed. We cannot be sure whether our estimated $w*$ values are a precise representation of the strength of uplift over water. Although $\Delta T$ does not provide realized uplift, it is an intuitively interpretable measure of uplift potential.

Uplift reduces the energetic costs of remaining airborne, for both soaring and flapping flyers. We used facultative soaring species as our model system. These species varied in morphological characteristics and soaring flight dependencies. Apart from minor differences in responding to wind support between ospreys and Eleonora's falcons, we found no significant species-specific variation in the impact of wind support and uplift on sea-crossing behaviour (electronic supplementary material, figure S1). This indicates that the patterns that we found can be true for sea-crossing in other species as well. Moreover, our study included all major bird migration flyways. As a result, our findings can further shed light on the energy seascapes other animals flying over these flyways would encounter. For example, the relatively high uplift during the night compared to daytime (figure 2) means less drag and could lead to energetically cheaper flight in nocturnal migrants over the Mediterranean Sea [54,55] and the Caribbean Sea [56,57] (but see [58] for a suggestion that songbirds prefer non-turbulent air for sea-crossing). Moreover, dragonflies [59] and cuckoos (*Cuculus* spp.) [60] migrate within the same time window as the Amur falcon over the Indian Ocean (figure 2), perhaps taking advantage of the energetic subsidy that the atmosphere provides.

## 5. Conclusion

The literature on migratory behaviour is increasingly recognizing that atmospheric conditions can reduce the risks associated with sea-crossing in terrestrial birds. Evidence of the role of wind support as a facilitator of this behaviour for many species is mounting. Yet, not much is known about the role of uplift. Our findings confirm the role of wind support as the main facilitator of over-water flight, but also provide evidence for widespread uplift potential over the open sea, at least in autumn. We provide quantitative evidence that $\Delta T$ is a meaningful proxy for uplift over the open sea and encourage future studies to take advantage of this proxy, not least because widespread use of $\Delta T$ will make comparisons among studies possible. Our findings suggest that the energetic costs of sea-crossing for soaring birds could be at least partially alleviated by overseas uplift. This may have important consequences for shaping routes, timing and strategies of birds crossing ecological barriers.

Ethics. Capture and tagging of ospreys in Europe were authorized by Istituto Nazionale per la Protezione e la Ricerca Ambientale and the National authority in Italy for Environmental Research and Protection.

The Ministry of Environment and Energy (Greece) granted permission for capturing and tagging Eleonora's falcons (licence nos. 6ΧΨΑ0-ΟΚΤ, Ψ9Θ24653Π8-PT3). The capture and tagging of Eleonora's falcons from Spain were authorized by the Regional Government of the Canary Islands (permit nos. ES-000844/2012, ES-000642/2013, 2014/2224, 2015/3835, 2017/6829). Oriental honey buzzards were tracked under permits no. 1705311 (Ministry of the Environment, Japan) and no. 6 (Aomori Prefecture). Capture and tagging of ospreys in North America were authorized by the USGS permit no. 20096 (USA) and scientific collecting permit 10907 (Canada). The work was approved by the University of North Carolina at Charlotte's IACUC, protocol 08–024, and by the IACUC of Drexel University, protocol 20632. The peregrine falcons in North America were captured under USGS permit no. 22749 and the state of New Jersey no. SC 2016120. Work on Peregrine falcons in Russian was done under permit no. 77-18/0854/4388 from The General Radio Frequency Centre, permit no. RU/2018/406 from Federal Service for Supervision of Communications, Information Technology and Mass Media (Roskomnadzor), and permit no. RU0000045099 from Federal Security Service. No specific permissions were required from Federal Service for Supervision of Natural Resources (Rosprirodnadzor) according to §44 and §6 of the Federal Law of the Russian Federation no. 52 from 24.04.1995 (last update 24.04.2020) 'On Wildlife', and from Federal Service for Technical and Export Control (FSTEC/FSTEK) according to Russian Federation government decree no. 633 from 29.08.2001 and Letter from FSTEK no. 240/33/1373 from 06.04.2015.

Data accessibility. All data used in this study are available from the Dryad Digital Repository: https://doi.org/10.5061/dryad.r4xgxd2ct [61]. R codes are available at https://github.com/mahle68/global_seascape_public.

Authors' contributions. E.N.: conceptualization, investigation, methodology, project administration, visualization, writing-original draft, writing-review and editing; G.B.: methodology, writing-review and editing; P.B.: project administration, writing-review and editing; R.O.B.: data curation, resources, writing-review and editing; O.D.: data curation, resources, writing-review and editing; J.F.: data curation, resources, writing-review and editing; L.G.: data curation, resources, writing-review and editing; S.G.: data curation, resources, writing-review and editing; H.H.: data curation, resources, writing-review and editing; C.K.: data curation, resources, writing-review and editing; O.K.: data curation, resources, writing-review and editing; N.L.: data curation, resources, writing-review and editing; F.M.: data curation, resources, writing-review and editing; I.P.: data curation, resources, writing-review and editing; A.S.: data curation, resources, writing-review and editing; J.-F.T.: data curation, resources, writing-review and editing; N.T.: data curation, resources, writing-review and editing; W.M.G.V.: data curation, resources, writing-review and editing; D.S.V.: data curation, resources, writing-review and editing; N.M.Y.: data curation, resources, writing-review and editing; M.W.: data curation, resources, writing-review and editing; K.S.: resources, validation, writing-review and editing

All authors gave final approval for publication and agreed to be held accountable for the work performed therein.

Competing interests. We declare we have no competing interests.

Funding. Tagging and tracking of Eleonora's falcons from Greece were conducted in the framework of the project 'LIFE13 NAT/GR/000909 Conservation measures to assist the adaptation of *Falco eleonorae* to climate change' with the financial support of the European Union LIFE Instrument and the Green Fund. Tracking of Eleonora's falcons from Spain was partly funded by the Cabildo de Lanzarote, European Social Fund, and adaptation and improvement of the internationalization of e-infrastructure of the ICTS-RBD for the ESFRI-LifeWatch. UvA-BiTS studies are facilitated by infrastructures for e-Ecology, developed with the support of LifeWatch, and conducted on the Dutch national e-infrastructure with the support of SURF Cooperative. Oriental honey buzzards were tracked as part of a project commissioned by the New Energy and Industrial Technology Development Organization (NEDO) in Japan. Tracking of ospreys in the Mediterranean was financially supported by the Foundation Prince Albert II de Monaco and the Associazone Italiana della Fondation Prince Albert II de Monaco ONLUS, the Corsica Natural Regional Park (France), the Maremma Regional Park Agency (Italy) and the Tuscan Archipelago National Park (Italy). P.B. was financially supported by

Erasmus+ ICM programme of the European Union and the University of Haifa. Development and maintenance of Env-DATA was in part supported by the US National Science Foundation award 1564380 and NASA award NNX11AP61G.

Acknowledgements. We thank all the people and institutions that supported fieldwork and data collection: A. Evangelidis, T. Fransson, T. Curk, P. Glazov, T. Nakahara, F. Nakayama, T. Suzumegano, D. Jennings, R. Julian, I. MacLeod, G. Hickey, J.J. Moreno, W. Bouten, M. de la Riva, J.L. Barroso, M. Majem, the "Progetto Falco pescatore" team, the protected areas of Maremma Regional Park, Tuscan Archipelago National Park, Diaccia Botrona Nature Reserve, WWF Orbetello Lagoon and Orti-Bottagone Nature Reserve in Italy, Massachusetts Audubon, New Jersey Audubon—Cape May Bird Observatory Center for Research and Education, Delaware State Parks, Newfoundland Power, Eversource Energy, the Jane B. Cook 1983 Charitable Trust, 3M, Squam Lakes Natural Science Center's Innovative Project Fund, and Meredith Bay Colony Club in USA, and the Canada Research Chair Program. We would like to thank Damien Farine, Michael Chimento, Emily Shepard, Adam Kane, and anonymous reviewers for their comments on the manuscript.

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
