## [Peer Review File · Proceedings of the Royal Society B: Biological Sciences]

Review History

RSPB-2021-0946.R0 (Original submission)

Review form: Reviewer 1

Recommendation

Reject – article is scientifically unsound

Scientific importance: Is the manuscript an original and important contribution to its field?

Excellent

General interest: Is the paper of sufficient general interest?

Good

Quality of the paper: Is the overall quality of the paper suitable?

Excellent

Is the length of the paper justified?

Yes

Should the paper be seen by a specialist statistical reviewer?

No

Do you have any concerns about statistical analyses in this paper? If so, please specify them explicitly in your report.

Yes

It is a condition of publication that authors make their supporting data, code and materials available - either as supplementary material or hosted in an external repository. Please rate, if applicable, the supporting data on the following criteria.

Is it accessible?

Yes

Is it clear?

Yes

Is it adequate?

Yes

Do you have any ethical concerns with this paper?

No

Comments to the Author

I reviewed a previous version of this paper for another journal. I generally liked it at the time and still like it now. The paper shows that uplift is widely available over the energy seascape, presents ΔT as a good proxy for uplift, and provides evidence that wind support seems to be the primary factor driving migrant route choice during water crossings. The latter is accomplished with a step selection function, which I think is a great application of the approach. I previously had major concerns about sample sizes for each species, especially in the context of drawing conclusions about soaring bird route selection on a “global” scale. Those have improved. I also had concerns about the number of steps used in the SSF for each individual; however, the authors have left that summary statistic out of table S1 in this version. Given that I know from the previous version that some of those sample sizes were incredibly small (e.g., 1), I need to see that those have improved to make a favorable recommendation for the paper.

Otherwise, I found the choice to subsample the species with larger sample sizes curious because the unevenness of the sample sizes among species shouldn’t bias the global estimates (you’re just throwing away information that could provide better estimates for those species).

Some line-by-line comments:

185-187 – you could use a continuous time model for the movement kernel and avoid doing this, *sensu* Eisaguirre et al. (2020, *J. Animal Ecology*).

189-191 – “over representation” shouldn’t be a problem. Including more data for a species will only improve the estimates for that species and would not bias the global estimates, given the hierarchical structure in the model explained on lines 210-211.

192-194 – Seems to be some conflicting info here. Did you fit gamma and von mises distributions to the empirical step lengths and turn angles and draw from those, or did you draw from the empirical distributions? Either is fine, just sort of unclear as written.

200-202 – could you include some justification for 40 years? Seems like individuals might choose their route based on the predictability of conditions along routes that they’ve experienced, rather than variability over 40 years. So, would it be better to make this covariate specific for each individual by taking the variance over the approximate age each individual?

Line 215-216 – did DIC and WAIC agree with CPO and MLik?

248-250 – I suggest consulting Avgar et al (2017, *Ecology and Evolution*) for some ideas on how

to better visualize and interpret this interaction.

251-259 – This is good discussion. I wonder if just the baseline uplift that the birds experience is enough, so they don't have to actively seek out stronger uplift to offset the energetic costs. Your point about turbulence is interesting as well. Perhaps there is some intermediate uplift strength (or ΔT) that is optimal, i.e., good uplift but not too turbulent. You could try testing this by including a quadratic ΔT term in the SSF.

268-269 – They could be more time-limited in the spring, however, so they may resort to flapping flight more in favor of a timely arrival on the breeding grounds rather than spending time seeking a more energetically optimal route.

Review form: Reviewer 2 (Adam Kane)

Recommendation

Accept with minor revision (please list in comments)

Scientific importance: Is the manuscript an original and important contribution to its field?

Good

General interest: Is the paper of sufficient general interest?

Excellent

Quality of the paper: Is the overall quality of the paper suitable?

Excellent

Is the length of the paper justified?

Yes

Should the paper be seen by a specialist statistical reviewer?

Yes

Do you have any concerns about statistical analyses in this paper? If so, please specify them explicitly in your report.

No

It is a condition of publication that authors make their supporting data, code and materials available - either as supplementary material or hosted in an external repository. Please rate, if applicable, the supporting data on the following criteria.

Is it accessible?

Yes

Is it clear?

Yes

Is it adequate?

Yes

Do you have any ethical concerns with this paper?

No

Comments to the Author

This is a well written and interesting movement ecology paper that looks at over-sea migration in terrestrial bird species. The authors go to some length in their methods to address how species

can afford these movements e.g. in assessing the viability of delta T as a measure of uplift. I have some minor comments throughout:

Abstract

Check spacing in the abstract, a few instances of words being squashed together. Indeed this occurs throughout the MS and may be a journal formatting issue.

Line 45 - I'd add the species names here if space permits.

Introduction

Bird morphology and body mass also have strong effects on flight costs too right.

Line 78 - "as well as based on morphological characteristics" sounds a bit awkward here. It's just that the behaviour varies according to flyway, season and morphology.

Line 87 - so this is when the sea surface is warmer than the air temperature.

Line 94 - stated that there are 5 species tested but in the abstract it's 4. (This seems to be clarified on line 181 where the grey-faced buzzard is excluded but you can appreciate the confusion).

Methods

On the route selection analysis are birds less selective of the energetically cheapest routes at different points of the journey? E.g. near the end of the crossing when land is in sight.

Line 190 - from not form.

Line 192 - Was it possible for one of these generated steps to send the bird over land?

Line 199 - It's worth defining 'wind support' here especially given it's importance in your findings.

Lines 212 - 213 - Presumably, these were 95% credible intervals? And I understand the analogy, but the terms aren't really significant in the same sense as a frequentist model.

Results and display items

All very clear and special mention to figure 2 which is a brilliant encapsulation of the work.

Why rely on a simple correlation when relating delta T to w^* when you have the GAM which shows a non-linear relationship?

Discussion

Good discussion on the value of the proxy delta T.

Supplementary Material

The supplementary figure legends have some question marks where they should refer to a specific table or figure.

Figure S1 needs to specify the random effects are based on varying slopes.

A minor point but the colour coding in S3 seems to be the opposite of what's natural to my mind - I'd think of orange as stronger than blue.

Please stretch the x axis for Figure S4, it's quite hard to distinguish among the different birds.

Great to see all of the code and data included.

Adam Kane

Decision letter (RSPB-2021-0946.R0)

07-Jun-2021

Dear Ms Nourani:

I am writing to inform you that your manuscript RSPB-2021-0946 entitled "The interplay of wind and uplift facilitates over-water flight in facultative soaring birds" has, in its current form, been rejected for publication in Proceedings B.

This action has been taken on the advice of referees, who have recommended that substantial revisions are necessary. With this in mind we would be happy to consider a resubmission, provided the comments of the referees are fully addressed. However please note that this is not a provisional acceptance. The reviewers and Associate Editor have some constructive critiques that will require some re-analysis of data.

Sincerely,
Dr John Hutchinson
mailto: proceedingsb@royalsociety.org

Associate Editor
Comments to Author:
Dear Dr. Nourani,

Thank you for submitting your work to PRSB. Two reviewers and myself have looked at the manuscript and found it quite interesting. There are however some issues with the data analysis and transparency in the presentation. As mentioned by reviewer #1 it is not necessary to remove data because the hierarchical structure in the model will automatically "weight" your estimates. I found the model selection to be inadequate and unnecessary. Removing variables with non-significant slopes is not a good practice. I really do not see the value of doing model selection here. You have a reasonable small model (few predictors) and a hierarchical structure that reflects the nature of your data. My recommendation would be to fit just one model (model 3) and report what you find.

Other comments:

Explain what data was used to fit the Gamma and von Mises. Also, did these distributions provided a good fit?

186. a tolerance of 30 minutes for hourly steps seems too high.

Priors and convergence diagnostics should be included.

It is a condition of publication that authors make the primary data, materials (such as statistical tools, protocols, software) and code publicly available.

Reviewer(s)' Comments to Author:

Referee: 1

Comments to the Author(s)

I reviewed a previous version of this paper for another journal. I generally liked it at the time and still like it now. The paper shows that uplift is widely available over the energy seascape, presents ΔT as a good proxy for uplift, and provides evidence that wind support seems to be the primary factor driving migrant route choice during water crossings. The latter is accomplished with a step selection function, which I think is a great application of the approach. I previously had major concerns about sample sizes for each species, especially in the context of drawing conclusions about soaring bird route selection on a "global" scale. Those have improved. I also had concerns about the number of steps used in the SSF for each individual; however, the authors have left that summary statistic out of table S1 in this version. Given that I know from the previous version that some of those sample sizes were incredibly small (e.g., 1), I need to see that those have improved to make a favorable recommendation for the paper.

Otherwise, I found the choice to subsample the species with larger sample sizes curious because the unevenness of the sample sizes among species shouldn't bias the global estimates (you're just throwing away information that could provide better estimates for those species).

Some line-by-line comments:

185-187 – you could use a continuous time model for the movement kernel and avoid doing this, *sensu* Eisaquirre et al. (2020, *J. Animal Ecology*).

189-191 – "over representation" shouldn't be a problem. Including more data for a species will only improve the estimates for that species and would not bias the global estimates, given the hierarchical structure in the model explained on lines 210-211.

192-194 – Seems to be some conflicting info here. Did you fit gamma and von mises distributions to the empirical step lengths and turn angles and draw from those, or did you draw from the empirical distributions? Either is fine, just sort of unclear as written.

200-202 – could you include some justification for 40 years? Seems like individuals might choose their route based on the predictability of conditions along routes that they've experienced, rather than variability over 40 years. So, would it be better to make this covariate specific for each individual by taking the variance over the approximate age each individual?

Line 215-216 – did DIC and WAIC agree with CPO and MLik?

248-250 – I suggest consulting Avgar et al (2017, *Ecology and Evolution*) for some ideas on how to better visualize and interpret this interaction.

251-259 – This is good discussion. I wonder if just the baseline uplift that the birds experience is enough, so they don't have to actively seek out stronger uplift to offset the energetic costs. Your point about turbulence is interesting as well. Perhaps there is some intermediate uplift strength (or ΔT) that is optimal, i.e., good uplift but not too turbulent. You could try testing this by including a quadratic ΔT term in the SSF.

268-269 – They could be more time-limited in the spring, however, so they may resort to flapping flight more in favor of a timely arrival on the breeding grounds rather than spending time seeking a more energetically optimal route.

Referee: 2

Comments to the Author(s)

This is a well written and interesting movement ecology paper that looks at over-sea migration in terrestrial bird species. The authors go to some length in their methods to address how species

can afford these movements e.g. in assessing the viability of delta T as a measure of uplift. I have some minor comments throughout:

Abstract

Check spacing in the abstract, a few instances of words being squashed together. Indeed this occurs throughout the MS and may be a journal formatting issue.

Line 45 - I'd add the species names here if space permits.

Introduction

Bird morphology and body mass also have strong effects on flight costs too right.

Line 78 - "as well as based on morphological characteristics" sounds a bit awkward here. It's just that the behaviour varies according to flyway, season and morphology.

Line 87 - so this is when the sea surface is warmer than the air temperature.

Line 94 - stated that there are 5 species tested but in the abstract it's 4. (This seems to be clarified on line 181 where the grey-faced buzzard is excluded but you can appreciate the confusion).

Methods

On the route selection analysis are birds less selective of the energetically cheapest routes at different points of the journey? E.g. near the end of the crossing when land is in sight.

Line 190 - from not form.

Line 192 - Was it possible for one of these generated steps to send the bird over land?

Line 199 - It's worth defining 'wind support' here especially given it's importance in your findings.

Lines 212 - 213 - Presumably, these were 95% credible intervals? And I understand the analogy, but the terms aren't really significant in the same sense as a frequentist model.

Results and display items

All very clear and special mention to figure 2 which is a brilliant encapsulation of the work.

Why rely on a simple correlation when relating delta T to w^* when you have the GAM which shows a non-linear relationship?

Discussion

Good discussion on the value of the proxy delta T.

Supplementary Material

The supplementary figure legends have some question marks where they should refer to a specific table or figure.

Figure S1 needs to specify the random effects are based on varying slopes.

A minor point but the colour coding in S3 seems to be the opposite of what's natural to my mind - I'd think of orange as stronger than blue.

Please stretch the x axis for Figure S4, it's quite hard to distinguish among the different birds.

Great to see all of the code and data included.

Adam Kane

Author's Response to Decision Letter for (RSPB-2021-0946.R0)

See Appendix A.

RSPB-2021-1603.R0

Review form: Reviewer 1

Recommendation

Accept with minor revision (please list in comments)

Scientific importance: Is the manuscript an original and important contribution to its field?

Excellent

General interest: Is the paper of sufficient general interest?

Excellent

Quality of the paper: Is the overall quality of the paper suitable?

Excellent

Is the length of the paper justified?

Yes

Should the paper be seen by a specialist statistical reviewer?

No

Do you have any concerns about statistical analyses in this paper? If so, please specify them explicitly in your report.

No

It is a condition of publication that authors make their supporting data, code and materials available - either as supplementary material or hosted in an external repository. Please rate, if applicable, the supporting data on the following criteria.

Is it accessible?

Yes

Is it clear?

Yes

Is it adequate?

Yes

Do you have any ethical concerns with this paper?

No

Comments to the Author

The authors did a great job of addressing my previous comments and concerns. I only have minor comments/suggestions below.

line 47: consider "greater uplift" instead of "higher uplift"

line 144: consider adding "The variable" at the start of the sentence (and maybe elsewhere too).

Not a big fan of starting sentences with symbols.

lines 202-204: consider rephrasing to reflect that you centered and scaled predictors to mean zero and unit variance.

lines 232-234: I'm assuming Fig. S1 is showing the species-level means compared to the global (all centered and scaled?). If that's the case, I suggest rewording to something like: "The model results suggested a greater preference for wind support in Eleonora's falcon and a weaker

preference in osprey (Fig. S1)." The term "significant" doesn't really hold the same meaning in Bayesian statistics as frequentist, so I think it's best to avoid it.

lines 244-249: nice addition, considering the new result

line 260: consider rewording to: "The map showing mostly positive deltaT indicates..."

figure 3 legend: see comment above regarding rephrasing instead of using "z-scores"

figure 4: nice way to show this interaction. sometimes these heat map representations of interactions are difficult to interpret, but I think it works well here.

supplementary figures: not seeing supp. figure legends anywhere. I may have just missed them, but please make sure they are included.

Decision letter (RSPB-2021-1603.R0)

13-Aug-2021

Dear Ms Nourani

I am pleased to inform you that your manuscript RSPB-2021-1603 entitled "The interplay of wind and uplift facilitates over-water flight in facultative soaring birds" has been accepted for publication in Proceedings B.

The referee(s) have recommended publication, but also suggest some minor revisions to your manuscript. Therefore, I invite you to respond to the referee(s)' comments and revise your manuscript. Because the schedule for publication is very tight, it is a condition of publication that you submit the revised version of your manuscript within 7 days. If you do not think you will be able to meet this date please let us know.

- 1) A text file of the manuscript (doc, txt, rtf or tex), including the references, tables (including captions) and figure captions. Please remove any tracked changes from the text before submission. PDF files are not an accepted format for the "Main Document".
- 2) A separate electronic file of each figure (tiff, EPS or print-quality PDF preferred). The format should be produced directly from original creation package, or original software format. PowerPoint files are not accepted.
- 3) Electronic supplementary material: this should be contained in a separate file and where possible, all ESM should be combined into a single file. All supplementary materials accompanying an accepted article will be treated as in their final form. They will be published alongside the paper on the journal website and posted on the online figshare repository. Files on

figshare will be made available approximately one week before the accompanying article so that the supplementary material can be attributed a unique DOI.

Sincerely,

Dr Maurine Neiman

Associate Editor

Comments to Author:

Dear Dr. Nourani,

Thank you for the revised version of your manuscript. As pointed out by reviewer #1, we cannot find the legends for the supplementary figures. Also, the reviewer has some good suggestion to improve your manuscript. Beyond those, consider the following:

line 132. Please explain the units used in this section.

line 173. Consider joining this paragraph with the previous one.

line 190. Where do these Gamma and von Mises distributions come from? Presumably fitted to the data?

line 210. I don't think is necessary to report how long it took the models to converge and on what machine. But, you should report how you assessed convergence.

Reviewer(s)' Comments to Author:

Referee: 1

Comments to the Author(s).

The authors did a great job of addressing my previous comments and concerns. I only have minor comments/suggestions below.

line 47: consider "greater uplift" instead of "higher uplift"

line 144: consider adding "The variable" at the start of the sentence (and maybe elsewhere too).

Not a big fan of starting sentences with symbols.

lines 202-204: consider rephrasing to reflect that you centered and scaled predictors to mean zero and unit variance.

lines 232-234: I'm assuming Fig. S1 is showing the species-level means compared to the global (all centered and scaled?). If that's the case, I suggest rewording to something like: "The model results suggested a greater preference for wind support in Eleonora's falcon and a weaker preference in osprey (Fig. S1)." The term "significant" doesn't really hold the same meaning in Bayesian statistics as frequentist, so I think it's best to avoid it.

lines 244-249: nice addition, considering the new result

line 260: consider rewording to: "The map showing mostly positive ΔT indicates..."

figure 3 legend: see comment above regarding rephrasing instead of using "z-scores"

figure 4: nice way to show this interaction. sometimes these heat map representations of interactions are difficult to interpret, but I think it works well here.

supplementary figures: not seeing supp. figure legends anywhere. I may have just missed them, but please make sure they are included.

Author's Response to Decision Letter for (RSPB-2021-1603.R0)

See Appendix B.

Decision letter (RSPB-2021-1603.R1)

16-Aug-2021

Dear Ms Nourani

I am pleased to inform you that your manuscript entitled "The interplay of wind and uplift facilitates over-water flight in facultative soaring birds" has been accepted for publication in Proceedings B.

Data Accessibility section

Open Access

Paper charges

Sincerely,

Proceedings B

Appendix A

Associate Editor

Comments to Author:

Dear Dr. Nourani,

Thank you for submitting your work to PRSB. Two reviewers and myself have looked at the manuscript and found it quite interesting. There are however some issues with the data analysis and transparency in the presentation. As mentioned by reviewer #1 it is not necessary to remove data because the hierarchical structure in the model will automatically "weight" your estimates. I found the model selection to be inadequate and unnecessary. Removing variables with non-significant slopes is not a good practice. I really do not see the value of doing model selection here. You have a reasonable small model (few predictors) and a hierarchical structure that reflects the nature of your data. My recommendation would be to fit just one model (model 3) and report what you find.

RE: We are grateful for all the constructive comments provided by the associate editor and the reviewers. We have now addressed all the concerns and comments, as detailed below. Regarding model selection, we agree with the associate editor and are now only reporting one model in the manuscript. We decided to go with model 2 instead of model 3, because we were interested to see whether and how the long-term variance of wind support can have an impact on route selection. With all the analyses done with this new version, we would like to point out that the results of the study remained largely unchanged.

Other comments:

Explain what data was used to fit the Gamma and von Mises. Also, did these distributions provided a good fit?

RE: We have now clarified that the step lengths and turning angles of over-water steps along the tracks were used to fit the distributions (separately for each species-flyway combination) on Ls. 198-199 (tracked-changes file). We investigated the diagnostic plots to assess the fit of the distributions and overall found a good fit.

186. a tolerance of 30 minutes for hourly steps seems too high.

RE: The tolerance of 30 minutes was chosen to maximize the amount of data that could be retained in the dataset (due to variation in sampling frequency of the original tracking data sets). We have now reduced the tolerance to 15 minutes. The number of data points is reduced, but not dramatically. Supplementary table S1 and modeling results are updated accordingly.

Priors and convergence diagnostics should be included.

RE: This information is now added to Ls. 217-220 (tracked-changes file).

It is a condition of publication that authors make the primary data, materials (such as statistical tools, protocols, software) and code publicly available.

RE: We have now uploaded all necessary data to reproduce the study on Dryad (<https://doi.org/10.5061/dryad.r4xgxd2ct>). Raw over-water tracking data, processed data used for step selection estimation, as well as any other data necessary to produce all the figures have been included. We have also updated all the R code necessary to reproduce the results and all the figures on Github (https://github.com/mahle68/global_seascape_public).

Reviewer(s)' Comments to Author:

Referee: 1

Comments to the Author(s)

I reviewed a previous version of this paper for another journal. I generally liked it at the time and still like it now. The paper shows that uplift is widely available over the energy seascape, presents ΔT as a good proxy for uplift, and provides evidence that wind support seems to be the primary factor driving migrant route choice during water crossings. The latter is accomplished with a step selection function, which I think is a great application of the approach. I previously had major concerns about sample sizes for each species, especially in the context of drawing conclusions about soaring bird route selection on a “global” scale. Those have improved. I also had concerns about the number of steps used in the SSF for each individual; however, the authors have left that summary statistic out of table S1 in this version. Given that I know from the previous version that some of those sample sizes were incredibly small (e.g., 1), I need to see that those have improved to make a favorable recommendation for the paper.

RE: We are happy to know that the reviewer finds the improvements that we made satisfactory and thank the reviewer for the insightful comments. The table that the reviewer is referring to (shown below) did not include the number of steps (the units in our model), but sea-crossing segments. We defined segments as parts of a sea-crossing track separated by islands. Each segment contains multiple steps, depending on its length. In the table below (from a previous version of the manuscript), we had one migratory track for the Peregrine falcon from North America and this track was not broken up by islands, so the whole track was one segment. We have since decided to remove this information about the number of segments from the manuscript, because 1) it can be easily misleading or misinterpreted, 2) does not provide any useful information, and 3) depends entirely on the geography of the area that the birds are flying over. However, as the reviewer has pointed out, we have since collected additional data, which is reflected in the new version of Table S1.

The number of steps per species used in the current version is as follows (note that this is highly dependent on the distance of sea-crossing, as well as the number of individuals and years of data available):

Eleonora's falcon (Greece): 10
Eleonora's falcon (Spain): 537
Osprey (the Americas): 1038
Osprey (Europe): 48
Oriental honey buzzard: 179
Peregrine falcon (the Americas): 27
Peregrine falcon (Europe): 19

Table S1 in the old version:

Table S1: Summary of biologging data used in the step-selection function analysis. The number of individuals (Individuals), number of migratory tracks (Tracks), and number of sea-crossing segments (Segments) correspond to the data that were retained after filtering the raw data set (see methods). The corresponding Movebank study names can be found in the footnotes.

Species	Tagging_location	Device	Years	Individuals	Tracks	Segments
Eleonora's falcon	Greece	GPS	2015	3	3	4
Oriental honey buzzard	Japan	GPS	2017-2018	8	14	32
Osprey ^a	South Europe	GPS	2013	3	4	6
Osprey ^b	North America	GPS	2009-2018	28	48	157
Peregrine falcon ^c	North Europe	GPS	2018	1	1	1
Peregrine falcon ^c	North America	GPS	2016	3	3	10
Totals				46	73	210

Here is Table S1 in the current version:

Table S1: Summary of bio-logging data used in the step-selection function analysis. The number of individuals and migratory tracks correspond to the data that were retained after filtering the raw data set (see methods). The corresponding Movebank study names are shown in the footnotes.

Species	Tagging_location	Device	Years	Individuals	Tracks
Eleonora's falcon	Greece	GPS	2015	3	3
Eleonora's falcon	Spain	GPS	2012-2019	19	38
Oriental honey buzzard	Japan	GPS	2017-2019	8	17
Osprey ^a	South Europe	GPS	2013-2019	4	5
Osprey ^b	North America	GPS	2009-2019	25	42
Peregrine falcon ^c	North Europe	GPS	2018-2020	4	5
Peregrine falcon ^c	North America	GPS	2016	2	2
Totals				65	112

Otherwise, I found the choice to subsample the species with larger sample sizes curious because the unevenness of the sample sizes among species shouldn't bias the global estimates (you're just throwing away information that could provide better estimates for those species).

RE: In the revised manuscript, we have forgone the sub-sampling step and have used all the data. The information in Table S1 and all the modeling results have been updated accordingly. The main results of the study remained unchanged, except that now we have a significant negative effect of the long-term variance of wind support on route selection.

Some line-by-line comments:

185-187 — you could use a continuous time model for the movement kernel and avoid doing this, sensu Eisaguirre et al. (2020, J. Animal Ecology).

RE: The method used by Eisaguirre et al. is interesting and promising, but our decision to use a one-hourly sub-sampled dataset was also to ensure that the resolution of the tracking data matches that of the environmental data (also one hourly). For this reason, we have kept the use of hourly tracking data in our analyses.

189-191 — “over representation” shouldn't be a problem. Including more data for a species will only improve the estimates for that species and would not bias the global estimates, given the hierarchical structure in the model explained on lines 210-211.

RE: We have now included all the data in our analysis, with no sub-sampling for individuals.

192-194 — Seems to be some conflicting info here. Did you fit gamma and von mises distributions to the empirical step lengths and turn angles and draw from those, or did you draw from the empirical distributions? Either is fine, just sort of unclear as written.

RE: We fit gamma and von mises distributions to the empirical step lengths and turning angles and drew from those. We have now clarified this in the manuscript (Ls. 198-199 in the tracked-changes file).

200-202 — could you include some justification for 40 years? Seems like individuals might choose their route based on the predictability of conditions along routes that they've experienced, rather than variability over 40 years. So, would it be better to make this covariate specific for each individual by taking the variance over the approximate age each individual?

RE: We used 40 years of data to calculate the long term conditions that populations from which these species are part of experience on average. We do not assume that the individual animals are choosing to reduce variability, but rather indirectly favor places characterized by lower (or higher) levels of overall variability, this can be based on genetic predisposition, social cues (following experienced individuals), etc. In order to show a shift in choice with experience would require fitting different step selection functions for each year of life (with the variance or average atmospheric conditions experienced in the previous years as predictors). This is definitely an interesting question to look into, but is outside the scope of our current study. Moreover, basing the decision to include multi-year variances on each individual's immediate response would make the assumption that flyways and places of good condition are subject to rapid changes and that there is a demographic change in preference (and selection criteria) as experience builds up. Finally, the choice of 40 years for the long-term conditions was due to the number of years for which ECMWF data is available (1979 to present).

Line 215-216 — did DIC and WAIC agree with CPO and Mlik?

RE: We had looked into WAIC and it agreed with the CPO and Mlik values. In the revised version of the manuscript, we have removed the model selection step, as suggested by the associate editor. We now only report the results for what was Model 2 in the previous version. Consequently, table 1 has been removed from the manuscript.

248-250 — I suggest consulting Avgar et al (2017, Ecology and Evolution) for some ideas on how to better visualize and interpret this interaction.

RE: We agree that there is a need to visualize and interpret the interaction term better. We consulted Avgar et al and found it a valuable resource. However, considering that our analyses followed Muff et al (2019, J Anim Ecol) to estimate the probability of use, we decided to visualize the interaction term as a raster image (Fig. 4) showing the probability of use predicted under different delta-t and wind support values.

251-259 — This is good discussion. I wonder if just the baseline uplift that the birds experience is enough, so they don't have to actively seek out stronger uplift to offset the energetic costs. Your point about turbulence is interesting as well. Perhaps there is some intermediate uplift strength (or deltaT) that is optimal, i.e., good uplift but not too turbulent. You could try testing this by including a quadratic deltaT term in the SSF.

RE: The question of how much uplift the birds need and the role of turbulence is definitely the next step in understanding the energetic of over-water flight in terrestrial birds. We explored the reviewer's suggestion by running the model with a smooth term for deltaT (using binned data and with no random effects for faster convergence) to see whether we find any signal for optimal conditions. The pattern is not very clear as shown in the effect plot below. We believe that addressing this issue will need high-resolution tracking (with GPS and ACC sensors) and higher resolution sea surface and air temperature data, to quantify vertical climb of the birds and to relate that to turbulence and delta-t that the bird experiences at finer spatio-temporal scales.

268-269 — They could be more time-limited in the spring, however, so they may resort to flapping flight more in favor of a timely arrival on the breeding grounds rather than spending time seeking a more energetically optimal route.

RE: This is a very good point. We have now pointed to seasonal differences in time- or energy-saving strategies in our discussion in L. 292 (tracked-changes file).

Referee: 2

Comments to the Author(s)

This is a well written and interesting movement ecology paper that looks at over-sea migration in terrestrial bird species. The authors go to some length in their methods to address how species can afford these movements e.g. in assessing the viability of delta T as a measure of uplift. I have some minor comments throughout:

RE: It's great to know that the reviewer finds the paper interesting. We have addressed the comments below and have checked and improved the formatting throughout.

Abstract

Check spacing in the abstract, a few instances of words being squashed together. Indeed this occurs throughout the MS and may be a journal formatting issue.

RE: We have now checked and fixed these problems.

Line 45 - I'd add the species names here if space permits.

RE: Unfortunately to stay within the required word limit, we had to skip naming the species.

Introduction

Bird morphology and body mass also have strong effects on flight costs too right.

RE: Bird morphology is definitely an important factor in determining flight costs. For this reason, in our introductory sentence, we say that atmospheric conditions “largely” define the energetic costs of flight to clarify that they are by no means the only determinant.

Line 78 – “as well as based on morphological characteristics” sounds a bit awkward here. It's just that the behaviour varies according to flyway, season and morphology.

RE: Done

Line 87 – so this is when the sea surface is warmer than the air temperature.

RE: This is true. We have now added this additional explanation in parentheses (L. 90 in the tracked-changes document).

Line 94 – stated that there are 5 species tested but in the abstract it's 4. (This seems to be clarified on line 181 where the grey-faced buzzard is excluded but you can appreciate the confusion).

RE: To reduce confusion, we changed the number of species from 4 to 5 in the abstract. We have collected the data and initiated the analysis for 5 species, but one is left out in the data preparation because of the low resolution. As the reviewer points out, this process has been explained in the methods section.

Methods

On the route selection analysis are birds less selective of the energetically cheapest routes at different points of the journey? E.g. near the end of the crossing when land is in sight.

RE: This is a very relevant question. The level of selectivity could be different at different stages of the journey. However, looking into this in the present study, which aims to reveal general patterns, is not very feasible because of sample size, but also because the species that we studied cross the sea at different lengths, in different points of the migratory journey, and in varying geographic conditions. We could have included a variable for distance to target in the model, for example, but due to the differences mentioned above, interpreting the coefficient of such a variable would be complicated. We think this question can best be answered when the focus of the study is a single species, then all the details of selectivity for energetically cheapest routes at different parts of the migratory journey could be investigated, and perhaps even compared between the seasons.

Line 190 - from not form.

RE: corrected

Line 192 - Was it possible for one of these generated steps to send the bird over land?

RE: We limited the alternative steps geographically so that all fall over water. It would be really informative to compare alternative steps over both land and water, but because the delta-t cannot be calculated over land, we limited the study to points over water only.

Line 199 – It's worth defining 'wind support' here especially given it's importance in your findings.

RE: We have defined wind support early on in the introduction (L. 82 in the tracked-changes document)

Lines 212 – 213 -Presumably, these were 95% credible intervals? And I understand the analogy, but the terms aren't really significant in the same sense as a frequentist model.

RE: Yes, we had considered the 95% credible intervals. The model selection step is now removed, as per the associate editor's suggestion. We now only present one model in the manuscript.

Results and display items

All very clear and special mention to figure 2 which is a brilliant encapsulation of the work.

Why rely on a simple correlation when relating delta T to w^* when you have the GAM which shows a non-linear relationship?

RE: We have now updated the methods and results sections to reflect the use of GAM to determine the relationship between w^* and delta T.

Discussion

Good discussion on the value of the proxy delta T.

RE: Thank you.

Supplementary Material

The supplementary figure legends have some question marks where they should refer to a specific table or figure.

RE: These have been corrected.

Figure S1 needs to specify the random effects are based on varying slopes.

RE: We have added this information now.

A minor point but the colour coding in S3 seems to be the opposite of what's natural to my mind – I'd think of orange as stronger than blue.

RE: The colors palette has been updated accordingly.

Please stretch the x axis for Figure S4, it's quite hard to distinguish among the different birds.

RE: We have improved the figure to make different species more distinguishable.

Great to see all of the code and data included.

RE: We appreciate the reviewer's support of our effort to have our results as reproducible as possible. We have now updated the code and data based on the revisions.

Appendix B

Associate Editor

Comments to Author:

Dear Dr. Nourani,

Thank you for the revised version of your manuscript. As pointed out by reviewer #1, we cannot find the legends for the supplementary figures. Also, the reviewer has some good suggestion to improve your manuscript. Beyond those, consider the following:

We are thankful to the editor and the reviewer for their positive comments and the swift review process. We have now incorporated the minor comments. Below is our response to the comments. We have now replaced the temporary URL with the permanent DOI of the Dryad repository associated with this manuscript. Regarding the issue with the legends of supplementary material, we have now uploaded all supplemental information as a single pdf file with all the legends included.

line 132. Please explain the units used in this section.

RE: We have included all the units in this section in square brackets (as standard abbreviations). We believe that this information will be sufficient for replicating the methodology.

line 173. Consider joining this paragraph with the previous one.

RE: Done

line 190. Where do these Gamma and von Mises distributions come from? Presumably fitted to the data?

RE: Yes, the distributions were fitted to the data. We have now clarified this.

line 210. I don't think is necessary to report how long it took the models to converge and on what machine. But, you should report how you assessed convergence.

RE: We used integrated nested Laplace approximation (INLA), which is a computationally less intensive alternative to MCMC. "INLA uses a combination of analytical approximations and numerical algorithms for sparse matrices to approximate the posterior distributions with closed-form expressions. This allows faster inference and avoids problems of sample convergence and mixing which permit to fit large datasets and explore alternative models." (Gómez-Rubio, 2020). Due to this benefit of using INLA, we did not have problems with convergence.

Reviewer(s)' Comments to Author:

Referee: 1

Comments to the Author(s).

The authors did a great job of addressing my previous comments and concerns. I only have minor comments/suggestions below.

line 47: consider "greater uplift" instead of "higher uplift"

RE: Done

line 144: consider adding "The variable" at the start of the sentence (and maybe elsewhere too). Not a big fan of starting sentences with symbols.

RE: Done

lines 202-204: consider rephrasing to reflect that you centered and scaled predictors to mean zero and unit variance.

RE: Done

lines 232-234: I'm assuming Fig. S1 is showing the species-level means compared to the global (all centered and scaled?). If that's the case, I suggest rewording to something like: "The model results suggested a greater preference for wind support in Eleonora's falcon and a weaker preference in osprey (Fig. S1)." The term "significant" doesn't really hold the same meaning in Bayesian statistics as frequentist, so I think it's best to avoid it.

RE: Done

lines 244-249: nice addition, considering the new result

RE: Thank you

line 260: consider rewording to: "The map showing mostly positive deltaT indicates..."

RE: Done

figure 3 legend: see comment above regarding rephrasing instead of using "z-scores"

RE: Done

figure 4: nice way to show this interaction. sometimes these heat map representations of interactions are difficult to interpret, but I think it works well here.

supplementary figures: not seeing supp. figure legends anywhere. I may have just missed them, but please make sure they are included.

RE: Thank you